# Cardioprotective Effects of Ursodeoxycholic Acid in Isoprenaline-Induced Myocardial Injury in Rats

**DOI:** 10.3390/biom14101214

**Published:** 2024-09-26

**Authors:** Dalibor Mihajlović, Đorđe Đukanović, Milica Gajić Bojić, Sanja Jovičić, Nebojša Mandić-Kovačević, Snežana Uletilović, Žana M. Maksimović, Nebojša Pavlović, Boris Dojčinović, Sergey Bolevich, Momir Mikov, Ranko Škrbić, Nada Banjac, Velibor Vasović

**Affiliations:** 1Emergency Department, Primary Healthcare Centre, 78000 Banja Luka, The Republic of Srpska, Bosnia and Herzegovina; 2Department of Emergency Medicine, Faculty of Medicine, University of Banja Luka, 78000 Banja Luka, The Republic of Srpska, Bosnia and Herzegovina; 3Centre for Biomedical Research, Faculty of Medicine, University of Banja Luka, 78000 Banja Luka, The Republic of Srpska, Bosnia and Herzegovinanebojsa.mandic-kovacevic@med.unibl.org (N.M.-K.); ranko.skrbic@med.unibl.org (R.Š.); 4Department of Pharmacy, Faculty of Medicine, University of Banja Luka, 78000 Banja Luka, The Republic of Srpska, Bosnia and Herzegovina; 5Department of Pharmacology, Toxicology and Clinical Pharmacology, Faculty of Medicine, University of Banja Luka, 78000 Banja Luka, The Republic of Srpska, Bosnia and Herzegovina; 6Department of Histology and Embryology, Faculty of Medicine, University of Banja Luka, 78 000 Banja Luka, The Republic of Srpska, Bosnia and Herzegovina; 7Department of Medical Biochemistry and Chemistry, Faculty of Medicine, University of Banja Luka, 78000 Banja Luka, The Republic of Srpska, Bosnia and Herzegovina; 8Department of Pharmacy, Faculty of Medicine, University of Novi Sad, 21000 Novi Sad, Serbia; 9Department of Pathologic Physiology, First Moscow State Medical University I.M. Sechenov, 119435 Moscow, Russia; 10Department of Pharmacology, Toxicology and Clinical Pharmacology, Faculty of Medicine, University of Novi Sad, 21101 Novi Sad, Serbia; momir.mikov@mf.uns.ac.rs (M.M.);; 11Academy of Sciences and Arts of the Republic of Srpska, 78000 Banja Luka, The Republic of Srpska, Bosnia and Herzegovina

**Keywords:** ursodeoxycholic acid, oxidative stress, isoprenaline-induced myocardial injury, cardioprotection

## Abstract

Patients suffering from cholelithiasis have an increased risk of developing cardiovascular complications, particularly ischemic myocardial disease. Ursodeoxycholic acid (UDCA), already used in clinical practice for the treatment of cholelithiasis and related conditions, has proven antioxidative, anti-inflammatory, and cytoprotective effects. Therefore, the aim of this study was to investigate the cardioprotective effect of UDCA pre-treatment on isoprenaline-induced myocardial injury in rats. Male Wistar albino rats were randomized into four groups. Animals were pre-treated for 10 days with propylene glycol + saline on days 9 and 10 (control), 10 days with propylene glycol + isoprenaline on days 9 and 10 (I group), 10 days with UDCA + saline on days 9 and 10 (UDCA group), and 10 days with UDCA + isoprenaline on days 9 and 10 (UDCA + I group). UDCA pre-treatment significantly reduced values of high-sensitivity troponin I (hsTnI) and aspartate aminotransferase (AST) cardiac markers (*p* < 0.001 and *p* < 0.01, respectively). The value of thiobarbituric acid reactive substances (TBARS) was also decreased in the UDCA + I group compared to the I group (*p* < 0.001). UDCA also significantly increased glutathione (GSH) levels, while showing a tendency to increase levels of superoxide dismutase (SOD) and catalase (CAT). The level of nuclear factor kappa-light-chain-enhancer of activated B cells (NF-κB) expression, a key regulatory gene of inflammation, was diminished when UDCA was administered. A reduction of cardiac damage was also observed in the UDCA pre-treated group. In conclusion, UDCA pre-treatment showed a cardioprotective effect on isoprenaline-induced myocardial injury in rats, primarily by reducing oxidative stress and inflammation.

## 1. Introduction

Cardioprotection is a commonly used term that includes all measures aimed at preventing heart injury and minimizing the consequences after cardiovascular events. Coronary heart disease is the most prevalent form of heart illness, often leading to acute myocardial infarction [1]. The prevalence of myocardial infarction is 3.8% in the general population and even higher, around 9.5%, among the elderly population, with expectations of further increase in the future [2]. The pharmacotherapy of myocardial infarction has improved over the years, although it still has areas that remain controversial [3,4]. Given the escalating prevalence and severity of myocardial injury, there is an urgent need to identify additional preventive and therapeutic measures for this condition.

Ursodeoxycholic acid (UDCA) is the most hydrophilic secondary bile acid normally present in the human bile pool, derived from the transformation of the primary bile acid chenodeoxycholic acid. UDCA has found wide application in the clinical treatment of cholestatic liver diseases [5]. Numerous studies have described the connection between cholelithiasis and coronary heart disease, reporting that patients suffering from cholelithiasis have an elevated risk of experiencing cardiovascular complications such as myocardial infarction [6]. Gallbladder and biliary diseases are generally associated with several heart conditions, including arrhythmias, myocardial ischemia, and Takotsubo syndrome [7]. Meta-analysis revealed that overall risk for developing cardiovascular condition is increased up to 33% among the patients with diagnosed gallstones [6]. Considering the link between these conditions and heart diseases, pre-treatment with UDCA requires careful evaluation of its potential impact on acute myocardial injury. The cytoprotective role of UDCA is well known, and its other biological effects, including anti-inflammatory and antioxidant effects, have also been reported [8,9,10].

For a long time, only emulsification of dietary fat was assigned as a role of bile acids. Nonetheless, growing evidence indicates that bile acids, including UDCA, can act as signaling molecules, regulating metabolism and immunity by modulating specific receptors. One of these receptors is Takeda G-protein receptor 5 (TGR5), which regulates the immune response and apoptosis and could minimize nuclear factor kappa-light-chain-enhancer of activated B cells (NF-κB) expression [11,12]. Activation of this receptor, alongside reducing the production of reactive oxygen species (ROS), may be a mechanism underlying the cytoprotective role of UDCA in acute myocardial injury [13]. In pre-clinical studies, it was shown that UDCA succeeded in reducing the infarct size and improving hemodynamic parameters in ischemia–reperfusion injury [14,15]. UDCA reduces arrhythmias caused by myocardial ischemia in rats and protects fetal cardiomyocytes against apoptosis in pregnant rats with intrahepatic cholestasis [16,17]. The combination of the above-mentioned beneficial properties suggests the potential for UDCA to be used to prevent and treat heart conditions, such as ischemic heart disease.

This study aimed to investigate the cardioprotective effects of UDCA on isoprenaline-induced myocardial injury in rats. Therefore, the effects of UDCA on cardiac biomarkers and oxidative stress markers were tested.

## 2. Materials and Methods

### 2.1. Experimental Animals

Male Wistar albino rats (*n* = 45) weighing 200 ± 20 g and aged 8 to 12 weeks, were kept under controlled laboratory conditions at room temperature (21 ± 2 °C), humidity (55 ± 5%), and a 12 h light–dark cycle. They had access to food and water ad libitum. At the beginning of the experiment, the animals were divided into four groups (Figure 1). Propylene glycol (PG) was used as the solvent for UDCA, whereas isoprenaline was dissolved in saline. UDCA or propylene glycol were administered by gavage for 10 days, while saline or isoprenaline were given subcutaneously on days 9 and 10. The dose of 25 mg/kg/day UDCA was selected based on a previous study demonstrating its safety and effectiveness over a 10-day period in reducing systemic and liver inflammation induced by lipopolysaccharide in rats [12]. Animals in the control group (C group) and the I group were pre-treated with propylene glycol (0.5 mL/kg) before receiving either saline (0.9% NaCl) or isoprenaline (85 mg/kg), respectively. In the UDCA and UDCA + I groups, animals were given UDCA (25 mg/kg) before receiving either saline (0.9% NaCl) or isoprenaline (85 mg/kg), respectively (Figure 1). Animals were anesthetized on the 11th day of the experiment using a combination of 90 mg/kg ketamine and 10 mg/kg xylazine and then sacrificed by exsanguination 24 h after the last administration of isoprenaline. Blood samples were collected from the inferior vena cava into serum separator tubes and plasma citrate tubes. The hearts were isolated and fixed in 10% formalin for further histological analyses.

The study on experimental animals, procedures, and protocols was approved by the Ethics Committee for the Protection and Welfare of Experimental Animals of the Faculty of Medicine, University of Banja Luka (number 18/1.51-15/21, dated 29 September 2021).

### 2.2. Serum Cardiac Markers and Lipid Profile Measurement

The concentrations of lipid parameters and liver enzymes that characterized heart damage in serum were measured on Abbott Alinity ci-series by chemiluminescence immunoassay, while the concentrations of high-sensitivity troponin I (hsTnI) and homocysteine (Hcy) were measured on Abbott Alinity ci-series by chemiluminescent microparticle immunoassay.

### 2.3. Heart Tissue Homogenization

Isolated hearts were washed with ice-cold saline and sliced; heart samples were then frozen and stored at −20 °C until homogenization. Heart homogenate was prepared in ice-cold phosphate buffer (pH 7.4) using an HG-15D homogenizer (Witeg Labortechnik GmbH, Wertheim, Germany) and then centrifuged at 4 °C and 1200× *g*. The supernatant was separated and used to determine the values of thiobarbituric acid reactive substances (TBARS), superoxide dismutase (SOD), catalase (CAT), and glutathione (GSH).

### 2.4. Oxidative Stress Markers

The index of lipid peroxidation, TBARS, was determined using 1% TBA and 0.05 mol/L sodium hydroxide (NaOH) and measured at 530 nm [18]. The activities of the antioxidative enzymes CAT and SOD, as well as the level of the antioxidant GSH, were analyzed using the Beutler methods and measured spectrophotometrically [19,20,21,22]. All parameters were determined in heart tissue homogenate.

### 2.5. Quantitative PCR Analysis

Heart tissue samples from experimental animals were stored in RNAlater^®^ RNA stabilization solution (Merck, St Louis, MO, USA) at −20 °C until RNA extraction. RNA was extracted from the heart tissue samples using the GenElute Mammalian Total RNA Miniprep Kit (Sigma-Aldrich, Germany), which includes all necessary reagents and silica membrane columns for total RNA isolation. After extraction, the concentration and purity of the isolated RNA was measured spectrophotometrically at a wavelength of 260 nm using a NanoDrop instrument (BioSpec-Nano, Shimadzu Scientific Instruments, Tokyo, Japan).

To quantify gene expression, mRNA must be converted into complementary DNA (cDNA) before performing polymerase chain reaction (PCR). The synthesis of cDNA from each RNA sample was carried out using the Enhanced Avian First Strand Synthesis Kit (Sigma-Aldrich, Darmstadt, Germany), following the manufacturer’s instructions. Gene expression values of interest were measured using quantitative real-time PCR with the Power SYBR Green PCR Master Mix kit (Applied Biosystems, Waltham, MA, USA). This kit contains Taq DNA polymerase, a mix of deoxynucleotide triphosphates, Mg^2+^, buffer, deionized sterile water, the SYBR Green fluorophore, and an inert ROX dye, whose fluorescence remains constant during the reaction, allowing normalization of SYBR Green fluorescence.

For each reaction, a melting curve profile was conducted. The quantitative values of the target genes were normalized on the expression of the housekeeping gene GAPDH. Primer sequences used for qRT-PCR amplification are given in Appendix A.

### 2.6. Histopathological Analysis

The collected samples of isolated hearts were shaped into blocks using paraffin wax and then sliced into 4 µm-thick sections using a standard microtome. These sections were subsequently treated with hematoxylin and eosin dye (H&E) to facilitate staining. Microscopic examination was conducted to assess myocardial injuries. Each slice was assigned a score ranging from 1 to 4, and an average score was computed for each group. A score of 1 indicated the absence of pathological changes in the myocardium, while a score of 2 denoted mild damage, characterized by scattered degeneration and minor infiltration of inflammation or focal damage to cardiomyocytes. Moderate damage, involving significant myofibril degeneration and/or widespread inflammation, was represented by a score of 3. A score of 4 represented severe damage, marked by necrosis coupled with diffuse inflammation.

### 2.7. Statistical Analysis

Statistical analysis was conducted using IBM-SPSS Statistics version 17.0 software (SPSS Inc., Chicago, IL, USA). The Kruskal–Wallis test was used to compare nonparametric traits among the groups. Tukey and Bonferroni tests were employed for post hoc analysis. Results are presented as mean ± standard error, with *p* < 0.05 considered statistically significant.

## 3. Results

### 3.1. Effects on Biochemical Parameters and Serum Cardiac Markers

The most sensitive marker of myocardial damage is cardiac troponin. The results showed that the hsTnI level was significantly increased in the isoprenaline (I) group compared to the control (C) group (*p* < 0.001), confirming that in isoprenaline-induced myocardial injury (MI) (Figure 2), UDCA pre-treatment prevents the increase of hsTnI induced by isoprenaline (*p* < 0.05).

There was a general statistical significance between the groups for aspartate aminotransferase (AST), alanine aminotransferase (ALT), and homocysteine (Hcy) levels (*p* < 0.05, *p* < 0.001, *p* < 0.001, respectively), with significantly higher values observed in the group that received isoprenaline compared to the control group (*p* < 0.01, *p* < 0.05, *p* < 0.05, respectively). UDCA significantly reduced the value of AST, which isoprenaline increased (*p* < 0.01) (Table 1).

### 3.2. Effects on Lipid Status

The lipid profile was analyzed to investigate the effects of UDCA on lipid metabolism. Overall statistical significance was found between the groups for low-density lipoprotein (LDL) and triglyceride (TG) values (*p* < 0.01 and *p* < 0.05, respectively) (Table 2). LDL and TG values were higher in isoprenaline groups (*p* < 0.05 for both). UDCA did not influence the lipid status in rats treated with isoprenaline.

### 3.3. Effects on Oxidative Stress Markers

Isoprenaline increased the value of TBARS, indicating ongoing oxidative stress (*p* < 0.001). UDCA pre-treatment prevents the oxidative stress caused by isoprenaline, which is reflected in reduction of TBARS (*p* < 0.001).

SOD, CAT, and GSH were analyzed as markers of antioxidative defense. There was a significant decrease in GSH values when isoprenaline was administered (Figure 3D). UDCA pre-treatment significantly increased GSH, strengthening antioxidative capacity (*p* < 0.01). This pre-treatment also showed a tendency towards increasing activities of SOD and CAT in heart tissue.

### 3.4. Effects on Gene Expression Levels

The expression levels of NF-κB and tumor necrosis factor-α (TNFα) genes were measured in heart tissue to evaluate the effect of UDCA on ongoing inflammatory processes in myocardial injury. The level of NF-κB gene expression was significantly higher when animals received isoprenaline (*p* < 0.05) (Figure 4). However, UDCA pre-treatment reduced this level in the UDCA + I group (*p* < 0.05). In the UDCA group, NF-κB gene expression was significantly lower compared to the C group. The level of TNFα gene expression was higher in the I group compared to the C group (*p* < 0.001). UDCA pretreatment did not attenuate the isoprenaline-induced increase of TNFα gene expression.

In heart tissue, the levels of Bcl-2-associated X protein (BAX) gene expression, a pro-apoptotic gene, were determined to evaluate apoptosis induction in myocardial injury. A higher level of BAX gene expression was observed in isoprenaline-treated animals (*p* < 0.001) (Figure 5). UDCA pre-treatment managed to reduce the level of BAX gene expression in the UDCA + I group compared to the I group (*p* < 0.001).

### 3.5. Pathohistological Analyses of Rat Hearts

Pathohistological analyses of heart samples of animals treated with isoprenaline revealed severely damaged myocardium with interstitial edema, infiltration of polymorphonuclear cells into the endomysium, and fragmented cardiomyocytes (Figure 6B). Pre-treatment with UDCA reduced the degree of myocardial damage caused by isoprenaline, showing mild intracellular cardiomyocyte edema and only initial fragmentation of cardiomyocytes. In animals that received UDCA pre-treatment, determined average myocardial injury score was lower compared to the I group (Figure 7).

## 4. Discussion

This study showed that pre-treatment with UDCA has a cardioprotective effect in an isoprenaline model of myocardial injury. A decrease in oxidative stress and a reduction in myocardial damage and serum cardiac markers were observed after ten days of pre-treatment with 25 mg/kg of UDCA. To the best of our knowledge, this study represents the first investigation into the effects of UDCA pre-treatment in the isoprenaline model of acute myocardial injury.

Myocardial injury induced by isoprenaline exhibits numerous metabolic and morphological aberrations in the heart tissue of experimental animals like those observed in human myocardial infarction [23]. Auto-oxidized intermediates of isoprenaline initiate lipid peroxidation, while isoprenaline overstimulation increases the level of NF-κB, both leading to cell death [24,25]. It is reported that even low doses of isoprenaline (32 μg/kg) result in abnormal histopathological changes as early as three hours post-injection, whereas moderate doses (85 mg/kg) lead to significant alterations in biochemical parameters and moderate cardiac necrosis [26,27]. To evaluate the cardioprotective effect of various natural and synthetic compounds, isoprenaline is often administered at a dose of 85 mg/kg over two consecutive days to induce myocardial infarction in rats [27,28,29]. During myocardial ischemia, the cellular membranes of cardiomyocytes rupture, resulting in the release of intracellular enzymes into the circulation. The degree of heart tissue damage correlates with the amount of enzyme released, with hsTnI, AST, and ALT being well-established serum markers of myocardial injury [30]. Among these markers, hsTnI is the most sensitive one, whose values were significantly elevated in animals treated with isoprenaline, as well as in Takotsubo syndrome [31,32,33]. An increase in transaminase levels, specifically AST and ALT, is characteristic of isoprenaline-induced myocardial injury [31,34]. In the present study, values of hsTnI, AST, and ALT were significantly higher in the group that received isoprenaline compared to the control group, indicating the presence of myocardial damage. A ten-day pre-treatment with UDCA alleviated these effects caused by isoprenaline, significantly reducing levels of hsTnI and AST, with a tendency to reduce ALT levels. It is important to consider that AST value better reflects events in the myocardium, while ALT is more specific to the liver [35]. This could be the reason for minor changes in ALT values. UDCA pre-treatment did not affect these parameters in the group that received saline instead of isoprenaline. This was expected, as UDCA’s cytoprotective effect only prevents the elevation of these parameters when damage is present.

An elevated concentration of homocysteine in plasma represents a risk factor for various cardiovascular diseases, including myocardial injury [36,37]. Homocysteine, in high concentrations, affects the activity of glutamate–cysteine ligase, an enzyme responsible for the de novo synthesis of glutathione [38]. Consequently, the concentration of glutathione decreases, leading to a reduced ability to scavenge reactive oxygen species (ROS), the onset of oxidative stress, and the worsening of myocardial injury. In our study, the administration of isoprenaline increased the value of homocysteine. These results are consistent with several other reports using the isoprenaline model of myocardial injury [39,40]. However, pre-treatment with UDCA did not have any effect on homocysteine serum values in our study. This issue needs to be additionally investigated, considering that there is only one report showing that UDCA can attenuate the homocysteine levels in an experimental model [12]. Importantly, UDCA has antioxidative potential that is crucial for reducing oxidative stress, thereby minimizing cardiac damage caused by isoprenaline.

Isoprenaline is a non-selective agonist of β-adrenergic receptors that acts via the cyclic adenosine monophosphate–protein kinase A (cAMP-PKA) pathway, initiating Ca^2+^ influx. This Ca^2+^ influx can further lead to ROS production, causing lipid peroxidation and myocardial infarction [41]. The most determined marker of lipid peroxidation in tissue or blood samples is TBARS [27,28,42]. UDCA reduces oxidative stress by scavenging ROS and boosting natural antioxidant defenses [13]. The results presented in this study showed that UDCA reduced the increase of TBARS in isoprenaline-treated animals. The activities of antioxidative enzymes, such as catalase and superoxide dismutase, along with the levels of GSH, enable the evaluation of antioxidative defense mechanisms. Superoxide radicals are transformed by SOD into hydrogen peroxide, which CAT then converts into molecular oxygen and water [43]. GSH serves as a substrate for glutathione peroxidase, aiding in the reduction of peroxide radicals [44]. A decrease in these three parameters is expected with the onset of oxidative stress. Oxidative stress partly contributes to myocardial damage induced by isoprenaline, and therefore, assessing the antioxidative capacity of heart tissue is crucial for evaluating the cardioprotective effects of certain compounds. Isoprenaline significantly decreases the value of GSH, which is restored with UDCA pre-treatment. Although there were no significant differences among the groups, a tendency towards decreased activities of SOD and CAT in the isoprenaline group, and their restoration after UDCA pre-treatment, can be observed. The elimination of hydrogen peroxide in heart mitochondria primarily occurs at the expense of GSH rather than of CAT [45]. This could explain the results, especially considering that the antioxidative capacity in our study was determined in heart tissue homogenate. In the group that did not receive isoprenaline, UDCA showed a tendency to boost the antioxidative capacity of heart tissue. This may be beneficial, as it could enhance the antioxidative defense response when myocardial injury occurs.

Maintaining a balanced lipid metabolism is crucial for cardiovascular health, as elevated levels of TC and LDL cholesterol are significant predictors of heart disease, closely associated with the higher incidence of acute myocardial infarction [46]. Isoprenaline increases cAMP levels, activating cAMP-dependent protein kinase, which phosphorylates hormone-sensitive lipase in fat cells [47]. This process initiates the breakdown of stored triacylglycerol, leading to hyperlipidemia. The lipid profile was assessed by measuring TC, LDL, HDL, and TG. In animals treated with isoprenaline, LDL and TG values were increased, while there was a tendency towards decrease in HDL value. These results are consistent with several other studies reporting isoprenaline’s impact on lipid profile [31,48]. UDCA is recognized for its ability to improve dyslipidemia [49]. However, in our study, UDCA pre-treatment did not significantly alter the lipid profile. This may be attributed to the short duration of UDCA application in our study, suggesting that longer exposure to UDCA could contribute to enhancing dyslipidemia.

Overstimulation with isoprenaline also initiates inflammatory processes in heart tissue. One of the most important markers of ongoing inflammation is the pro-inflammatory cytokine TNFα. After myocardial injury, activated NF-κB released from cardiomyocytes induces the transcription of pro-inflammatory genes, including TNFα [25,50]. Therefore, in pro-inflammatory signaling pathways, NF-κB is considered a key regulator. Upregulation of NF-κB and TNFα genes was observed in animals that received isoprenaline in our study. Several other studies have also showed similar effects of isoprenaline on pro-inflammatory gene expression [25,51,52]. There is evidence that bile acids, including UDCA, act as signaling molecules that regulate immunity and metabolism through the activation of the membrane TGR5 [11,53]. The transcription of NF-κB and its following effects can be antagonized by the activation of TGR5 [54,55]. Indeed, UDCA pre-treatment caused downregulation of NF-κB in isoprenaline-treated rats, with a tendency towards downregulation of the TNFα gene. Downregulation of NF-κB and TNFα caused by UDCA has already been described by several authors [12,55,56]. In the UDCA group, NF-κB was significantly downregulated, indicating a strong inhibitory effect of UDCA. Given that UDCA is a TGR5 agonist, this pronounced downregulation of NF-κB is an expected result of TGR5 activation. However, the exact mechanisms of UDCA involved in the regulation of inflammatory processes in heart tissue after myocardial injury need to be further investigated.

Apoptosis is a critical pathological feature in acute myocardial infarction and heart failure, with animal studies showing that its inhibition has beneficial effects [57]. The myocardial injury caused in the isoprenaline model is associated with increased apoptotic activity [58]. BAX is the key pro-apoptotic protein involved in the programmed cell death of cardiomyocytes [59]. The results of this study showed that isoprenaline increases the BAX gene expression in heart tissue. Other studies also support these results, showing upregulation of BAX in isoprenaline-treated animals [25,60]. In our study, it was demonstrated that the increased expression of the BAX gene was diminished by UDCA pre-treatment. These findings are consistent with previous reports that showed BAX downregulation caused by UDCA, simultaneously minimizing onset of apoptosis [56]. As expected, UDCA itself did not alter BAX gene expression in the absence of isoprenaline administration. A positive correlation between NF-κB and BAX suggests the involvement of the NF-κB pathway in the apoptosis of cardiomyocytes [59]. Our study showed a positive correlation in the upregulation of NF-κB and BAX in isoprenaline and their downregulation when UDCA was given to animals in pre-treatment.

High doses of isoprenaline, such as 85 mg/kg, have been previously reported to cause diffuse heart injury [61]. Prior to fixing the heart in 4% formaldehyde, the tissue was examined for macroscopic signs of damage, including changes in color or thickness in specific regions of the heart muscle. It was observed that all components of the heart were uniformly enlarged, with diffuse heart injury present. In our previous study using the isoprenaline model, right ventricular wall thickness and cardiomyocyte size were measured, and it was concluded that the increased thickness resulted from cardiomyocyte hypertrophy and interstitial edema [58]. Consequently, triphenyl tetrazolium chloride (TTC) staining to assess the ischemic zone was not performed in this study. Pathohistological examination revealed that the control group’s cardiac tissue maintained a normal homogeneous structure, indicating no myocardial damage. Isoprenaline caused severe heart damage, characterized by fragmented and degenerated cardiomyocytes and a dense inflammatory infiltrate composed of polymorphonuclear cells. Similar pathohistological findings in cardiac tissue have been described in rats exposed to isoprenaline [27,28]. The results from this study showed that UDCA pre-treatment protects cardiomyocytes and heart tissue from isoprenaline-induced damage. This was also observed in the lowering of the average myocardial damage score, supporting the cardioprotective effect of UDCA.

UDCA can regulate various pathways and receptors. Its cardioprotective effects against hypoxia are known to involve the modulation of multiple receptors and transcriptional mediators of cellular stress, such as hypoxia-inducible factor 1α (HIF-1α) and p53 protein [61,62]. Similarly, in our study, the prevention of myocardial injury caused by isoprenaline using UDCA also involves multiple mechanisms.

## 5. Conclusions

The present study demonstrated that pre-treatment with 25 mg/kg of UDCA has beneficial effects in preventing myocardial infarction induced by isoprenaline. The advantageous effects of UDCA are reflected in the decrease in cardiac injury markers and oxidative stress, and improvement in antioxidative defense, as well as in the downregulation of pro-inflammatory and pro-apoptotic genes. Based on our findings, it could be expected that cholestatic patients treated with UDCA may experience better outcomes in the event of an ischemic myocardial injury. Nevertheless, further investigations are needed to confirm these findings in humans, as well as to thoroughly describe the molecular mechanisms behind the cardioprotective effects of UDCA.

## Figures and Tables

**Figure 1 biomolecules-14-01214-f001:**
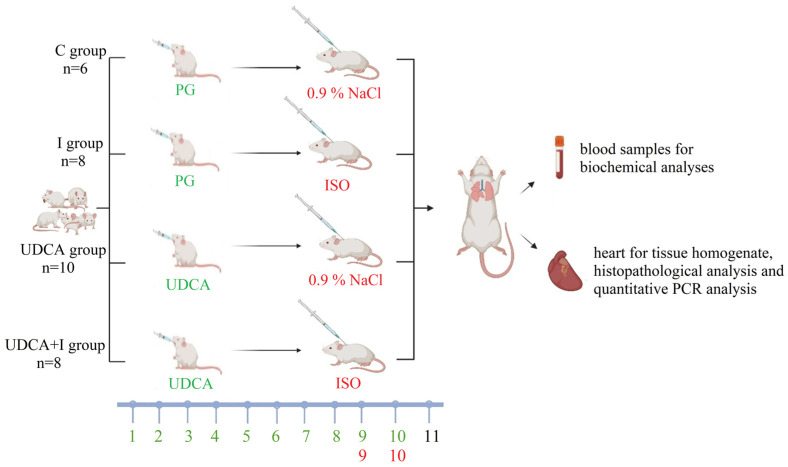
Graphical scheme of experimental groups and protocol. PG—propylene glycol; ISO—isoprenaline; UDCA—ursodeoxycholic acid; C—control group; I—isoprenaline-treated group; UDCA group—ursodeoxycholic-acid- and saline-treated group; UDCA+I—ursodeoxycholic-acid- and isoprenaline-treated group.

**Figure 2 biomolecules-14-01214-f002:**
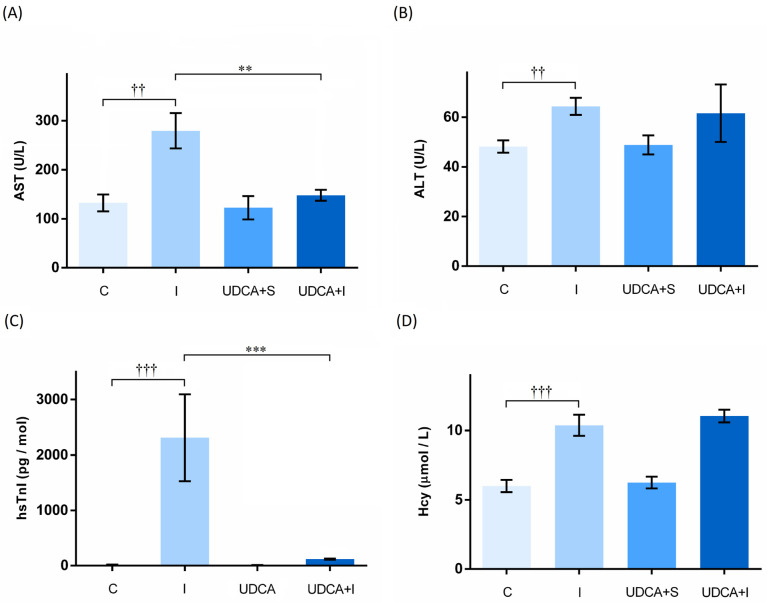
Effects of UDCA pre-treatment on serum biochemical parameters. (**A**)—Effect on aspartate aminotransferase (AST); (**B**)—Effect on alanine aminotransferase (ALT); (**C**)—Effect on high-sensitivity troponin I (hsTnI); (**D**)—Effect on homocysteine (Hcy); Values are expressed as mean ± SEM. C—control group; I—isoprenaline group; UDCA—ursodeoxycholic-acid- and saline-treated group; UDCA + I—ursodeoxycholic-acid- and isoprenaline-treated group. Asterisk (*) indicates significant differences between the UDCA + I and I group, ** *p* < 0.01, *** *p* < 0.001, and sign (†) indicates significant differences between the I and C group, †† *p* < 0.01, ††† *p* < 0.001.

**Figure 3 biomolecules-14-01214-f003:**
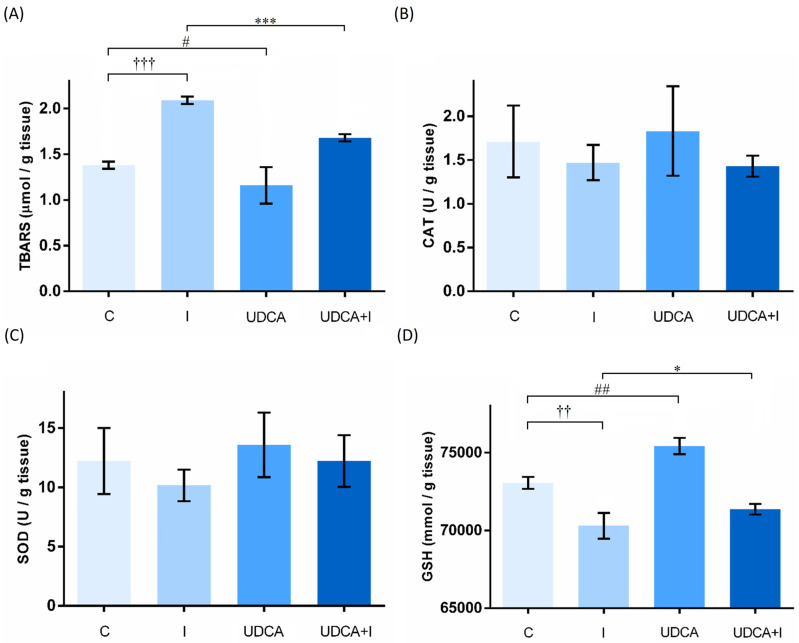
Effects of UDCA pre-treatment on heart tissue homogenate thiobarbituric acid reactive substances (TBARS) (**A**), antioxidative enzyme levels—catalase (CAT) (**B**) and superoxide dismutase (SOD) (**C**), and reduced glutathione (GSH) (**D**). Data are expressed as mean ± SE. C—control group; I—isoprenaline-treated group; UDCA—ursodeoxycholic-acid- and saline-treated group; UDCA + I—ursodeoxycholic-acid- and isoprenaline-treated group. One-way ANOVA, Bonferroni test, and Mann–Whitney test were performed, and asterisk (*) indicates significant differences between the UDCA + I and I group; * *p* < 0.05, *** *p* < 0.001, and sign (†) indicates significant differences between the I and C group; †† *p* < 0.01; ††† *p* < 0.001; hashtag (#) indicates significant differences between the UDCA and C group; # *p* < 0.05; ## *p* < 0.01.

**Figure 4 biomolecules-14-01214-f004:**
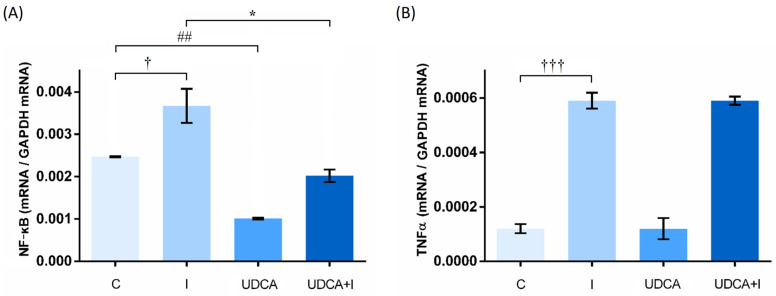
Effects of UDCA pre-treatment on NF-κB (**A**) and TNFα (**B**) gene expression in heart tissue. Data are expressed as mean ± SE. C—control group; I—isoprenaline-treated group; UDCA—ursodeoxycholic-acid- and saline-treated group; UDCA + I—ursodeoxycholic-acid- and isoprenaline-treated group. One-way ANOVA, Bonferroni test, and Mann–Whitney test were performed and asterisk (*) indicates significant differences between the UDCA + I and I group; * *p* < 0.05 and sign (†) indicates significant differences between the I and C group; † *p* < 0.05; ††† *p* < 0.001; hashtag (#) indicates significant differences between the UDCA and C group; ## *p* < 0.01.

**Figure 5 biomolecules-14-01214-f005:**
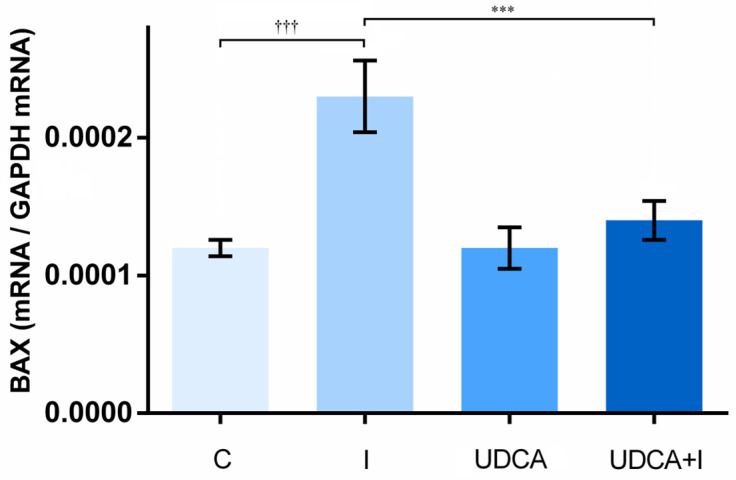
Effects of UDCA pre-treatment on BAX gene expression in heart tissue. Data are expressed as mean ± SE. C—control group; I—isoprenaline-treated group; UDCA—ursodeoxycholic-acid- and saline-treated group; UDCA + I—ursodeoxycholic-acid- and isoprenaline-treated group. One-way ANOVA, Bonferroni test, and Mann–Whitney test were performed, and asterisk (*) indicates significant differences compared with the I group; *** *p* < 0.001, and sign (†) indicates significant differences compared with the C group; ††† *p* < 0.001.

**Figure 6 biomolecules-14-01214-f006:**
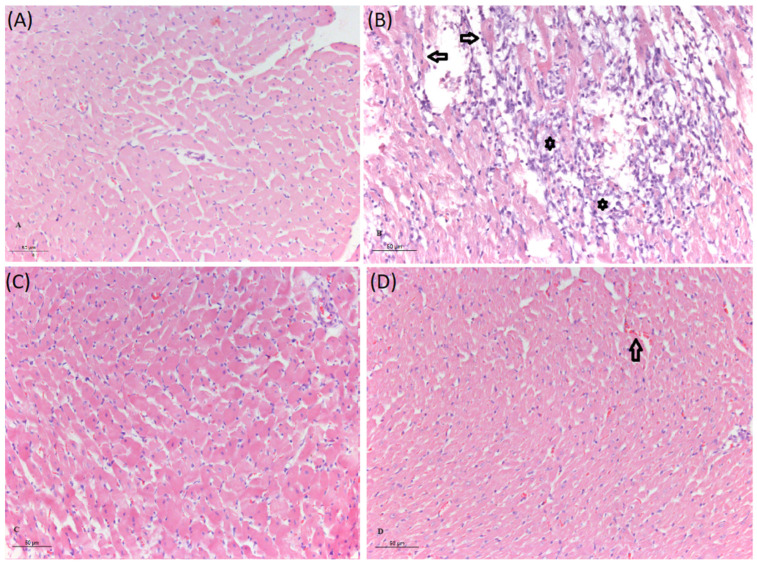
Histopathological section of rat heart, H&E staining, magnification ×20, scale bar 50 µm. The C group (**A**) shows normal histological structure of the heart muscle. The I group (**B**) shows focal infiltration of the endomysium with polymorphonuclear cells (star), and fragmentation of cardiomyocytes (arrow). The UDCA group (**C**) shows normal histological structure of the heart muscle. The UDCA + I group (**D**) shows mild intracellular cardiomyocyte edema with initial fragmentation and scarce extravasation of erythrocytes (arrow).

**Figure 7 biomolecules-14-01214-f007:**
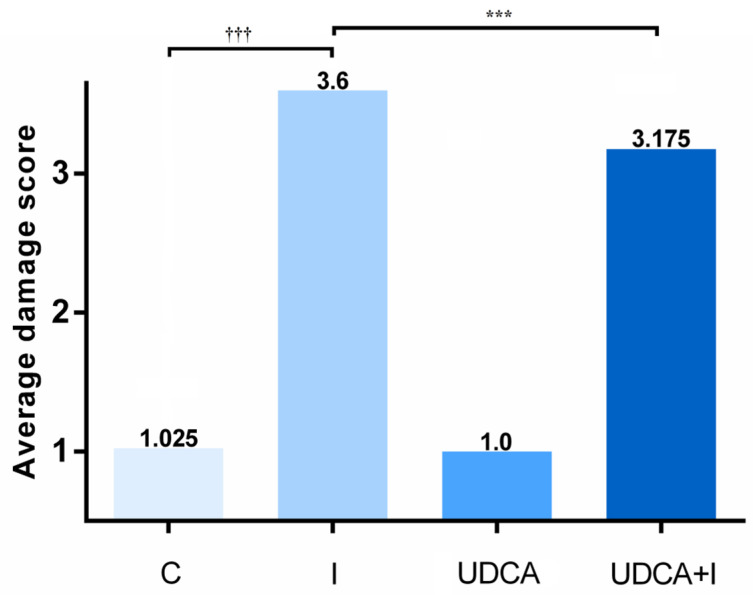
Effects of UDCA pre-treatment on myocardial damage score. The values represent the average damage score of the group. The following scoring system was used: score 1: no changes, normal histological finding; score 2: mild damage (focal damage of the myocardium or multifocal degeneration of myocardocytes with mild infiltration); score 3: moderate damage (pronounced myofibrillar degeneration and/or diffuse inflammatory process); and score 4: significant damage (necrosis with inflammatory process). C—control group; I—isoprenaline-treated group; UDCA—ursodeoxycholic-acid- and saline-treated group; UDCA + I—ursodeoxycholic-acid- and isoprenaline-treated group. The Kruskal–Wallis test was used to compare nonparametric traits among the groups, and asterisk (*) indicates significant differences compared with the I group; *** *p* < 0.001, and sign (†) indicates significant differences compared with the C group; ††† *p* < 0.001.

**Table 1 biomolecules-14-01214-t001:** Effect of UDCA pre-treatment on serum biochemical parameters.

	Groups (Mean ± SD)			
Parameters	C	I	UDCA	UDCA + I
AST (U/L)	132.33 ± 21.56	279.38 ± 101.87 ††	122.51 ± 34.12	148.01 ± 35.48 **
ALT (U/L)	48.17 ± 6.05	64.38 ± 9.74 ††	48.83 ± 9.41	61.60 ± 11.73
hsTnI (pg/mol)	14.18 ± 10.96	2310.88 ± 2141.70 †††	6.50 ± 5.71	116.85 ± 33.68 ***
Hcy (μmol/L)	6.00 ± 1.08	10.38 ± 2.16 †††	6.25 ± 1.05	11.05 ± 1.4

Data are expressed as mean ± SD. AST—aspartate aminotransferase; ALT—alanine aminotransferase; Hcy—homocysteine; hsTnI—high-sensitivity troponin I; C—control group; I—isoprenaline-treated group; UDCA—ursodeoxycholic-acid- and saline-treated group; UDCA + I—ursodeoxycholic-acid- and isoprenaline-treated group. One-way ANOVA, Bonferroni test, and Mann–Whitney test were performed; asterisk (*) indicates significant differences compared with the I group, ** *p* < 0.01, *** *p* < 0.001, and sign (†) indicates significant differences compared with the C group †† *p* < 0.01, ††† *p* < 0.001.

**Table 2 biomolecules-14-01214-t002:** Effect of UDCA pre-treatment on lipid status.

	Groups (Mean ± SD)			
Parameters	C	I	UDCA	UDCA + I
TC (mmol/L)	1.78 ± 0.35	1.86 ± 0.23	1.88 ± 0.21	1.71 ± 0.17
LDL (mmol/L)	0.27 ± 0.05	0.44 ± 0.12 ††	0.23 ± 0.05	0.36 ± 0.07
HDL (mmol/L)	0.57 ± 0.05	0.55 ± 0.08	0.63 ± 0.05	0.55 ± 0.05
TG (mmol/L)	0.48 ± 0.08	0.84 ± 0.24 ††	0.74 ± 0.73	0.81 ± 0.08

Data are expressed as mean ± SD. TC—total cholesterol; LDL—low-density lipoprotein cholesterol; HDL—high-density lipoprotein cholesterol; TG—triglycerides; C—control group; I—isoprenaline-treated group; UDCA—ursodeoxycholic-acid- and saline-treated group; UDCA + I—ursodeoxycholic-acid- and isoprenaline-treated group. One-way ANOVA, Bonferroni test, and Mann–Whitney test were done, and sign (†) indicates significant differences compared with the C group; †† *p* < 0.01.

## Data Availability

The data presented in this study are available on request from the corresponding author.

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
