# Peer review of "Cardioprotective Effects of Ursodeoxycholic Acid in Isoprenaline-Induced Myocardial Injury in Rats"

_biomolecules, 2024, doi:10.3390/biom14101214_

Round 1

Reviewer 1 Report

Comments and Suggestions for Authors

The authors investigated the preventive effects of UCDA in a model of isoproterenol-induced myocardial infarction. The add-on value of the presented results versus the current knowledge is not clear. The relevance of the subject needs to be better emphasized. Finally, several issues need to be solved.

The introduction section indicates that previous reports have already reported cardioprotective effects of UDCA against myocardial infarction. What does the present study add to the current knowledge?

Introduction section: the paragraph regarding isoproterenol should be deleted. It does not add significative value to the introduction section. The only concern is the justification of the model which should be discussed in the Discussion section instead.

Groups description, please add a schematic of the protocol, it would help the reader. Please also clarify the sentences (lines 99 to 105). It is clearer in the abstract. Please also confirm that the vehicle for UCDA was propylene glycol.

Please indicate the delay between the last administration of isoproterenol and exsanguination.

The study should also include histological examination or at least TTC evaluation. It would reinforce the semi-quantitative histopathological examination. In addition, representative pictures would be very interesting as the effect based on hsTnI measurements appears striking.

There is a discrepancy between the results presented in Figure 6 and those shown in Table 1. How do you explain the damage score of UDCA + I close to the I group while hs TNI shows a dramatic reduction in hsTnI in the UDCA + I group? Once again, TTC analysis would be useful, assessement of left venticular function also. Finally, there is no symbol in Figure 6 and non-parametric test should be used (not indicated in the statistical analysis paragraph) although a difference is stated.

Lines 228-230: please delete if the difference is not statistically significant.

What is the rationale for choosing the dose of UCDA. Please indicate how it is relevant to human treatment?

Reviewer 2 Report

Comments and Suggestions for Authors

Please find the attached PDF for comments to the authors.

Comments on the Quality of English Language

 Minor editing of English language required.

Reviewer 3 Report

Comments and Suggestions for Authors

In this submission, the authors explore the cardioprotective effect of ursodeoxycholic acid (UDCA) in myocardial infarction induced by isoprenaline injection.  The authors pretreated the mice with either UDCA or propylene glycol. After the isoprenaline injection, the mice were sacrificed and assayed using several techniques to assess cardiac damage and inflammation.

The scientific findings are sound, and multiple lines of evidence back up their conclusions; however, the paper is difficult to read, and several errors in the text need to be corrected.   I have one primary concern and several minor concerns.

Major concern:

There are no proper controls in which mice are completely untreated before isoprenaline injection, so we cannot evaluate if propylene glycol has an effect.

Minor concerns:

Throughout the text, abbreviations are used without definition, making the text difficult to read.  Please define each abbreviation at its first usage in the text.

I assume propylene glycol is the solvent for UDCA, but this needs to be explicitly stated in the text.

The significance of each of the markers used needs to be explained more.  Please explain in the text why you are analyzing specific markers and their individual significance.

In instances where there is no statistical significance please clarify if this finding supports you hypothesis.  For example serum AST was not impacted by UDCA treatment, does this matter or impact your hypothesis?

Statistical analysis needs to be completed. Please compare each group, not just the Iso and the Iso+ UDCA groups. For figures 1, 2,3 and 5  use a line to indicate which groups are compared to which other groups.  Are the groups only being compared to the control?

 In Figure 1, please chart all the measures with statistics, not just troponin.

In Figure 3, it appears that the UDCA group is statistically different from all the other groups in the analysis of NFkB, but it is not marked so. This could impact your interpretation.

Figure 4 should be removed since these data are repeated identically in Figure 5.

Round 2

Reviewer 1 Report

Comments and Suggestions for Authors

Thanks to the authors for their answers. I have 2 remaining comments :

1- (see Comment/Response 5), please add some discussion (and Ref if needed) in the manuscript regarding the use of TTC.

2- (see Comment/Reponse 8), please indicate in the methods some rationale (and Ref if needed) for choosing the dose of UCDA.

Reviewer 2 Report

Comments and Suggestions for Authors

The Authors addressed all my comments appropriately.

Comments on the Quality of English Language

Minor editing/proof reading of English language may be required.

Reviewer 3 Report

Comments and Suggestions for Authors

The revised work corrects the many minor errors of the first draft and provides support for their use of this particular system without a no-pretreatment control, which was my main concern about the text.

The work and its premise are well-described.
